# UV-Nanoimprint Lithography for Predefined SERS Nanopatterns Which Are Reproducible at Low Cost and High Throughput

**DOI:** 10.3390/nano13101598

**Published:** 2023-05-10

**Authors:** Karolina Milenko, Firehun Tsige Dullo, Paul C. V. Thrane, Zeljko Skokic, Christopher A. Dirdal

**Affiliations:** SINTEF Microsystems and Nanotechnology, Gaustadalleen 23C, 0737 Oslo, Norway; paul.thrane@sintef.no (P.C.V.T.); zeljko.skokic@sintef.no (Z.S.); christopher.dirdal@sintef.no (C.A.D.)

**Keywords:** surface-enhanced Raman scattering (SERS), Raman spectroscopy, UV-nanoimprint lithography (UV-NIL), nanopatterning, sensors

## Abstract

A controlled and reliable nanostructured metallic substrate is a prerequisite for developing effective surface-enhanced Raman scattering (SERS) spectroscopy techniques. In this study, we present a novel SERS platform fabricated using ultra-violet nanoimprint lithography (UV-NIL) to produce large-area, ordered nanostructured arrays. By using UV-NIL imprinted patterns in resist, we were able to overcome the main limitations present in most common SERS platforms, such as nonuniformity, nonreproducibility, low throughput, and high cost. We simulated and fabricated C-shaped plasmonic nanostructures that exhibit high signal enhancement at an excitation wavelength of 785 nm. The substrates were fabricated by directly coating the imprinted resist with a thin gold layer. Avoiding the need to etch patterns in silicon significantly reduces the time and cost of fabrication and facilitates reproducibility. The functionality of the substrates for SERS detection was validated by measuring the SERS spectra of Rhodamine 6G.

## 1. Introduction

Surface-enhanced Raman scattering (SERS) is a very powerful and promising technique, since it allows for the “fingerprint” of specific molecules to be obtained without the need for labelling elements [1]. Compared to well-established methods, such as high-performance liquid chromatography (HPLC) or gas chromatography–mass spectrometry (GC-MS), SERS can avoid the complicated and time-consuming sample pretreatment, allowing for real-time detection. Moreover, SERS exhibits less interference from water and, therefore, could be used in a wide range of applications, such as environmental detection [2], food safety [3], and biomedical diagnostics [4]. However, one limitation of SERS that is often addressed in the literature is the reproducibility of the results [5]. Recently, more and more publications refer to the gap between the high quality and high enhancement presented in publications and the relatively limited number of practical applications [6,7,8,9]. The challenges of SERS as regards out-of-the-lab applications are related to the homogeneity and reproducibility of the SERS signal, which is often related to variations in the commonly used colloidal nanoparticles (NPs), where the aggregation is difficult to control and, therefore, the enhancement is difficult to reproduce on a large scale. Recently, a study of six commercially available SERS substrates revealed relative standard deviations ranging from 2.7% to about 50–60% depending on the SERS substrate, wavelength, and microscope objective used [10]. Additionally, the shape and size of the SERS nanopattern has an influence on the obtained enhancement and, therefore, for maximum enhancement, it should be specifically designed for the wavelength and sample being tested [11,12,13]. Electron-beam lithography (EBL) and focused ion beam (FIB) milling offer excellent resolution and can be used to fabricate a wide variety of desired nanopatterns with good uniformity and reproducibility [14]; however, it is costly and time-consuming owing to it being a direct-writing process. Nanosphere lithography has been proposed as an EBL-free fabrication method [15,16,17]. A monolayer of close-packed polystyrene colloid microspheres is used as a template for the preparation of nanostructured arrays. Although structures fabricated with this technique show good enhancement, only a limited number of patterns, such as nanotriangles or film over nanospheres, can be fabricated. In recent years, nanoimprint lithography (NIL) has been investigated as a method for fabricating plasmonic substrates [18,19,20,21]. Originally, the NIL imprinted resists were used as a mask to etch nanostructures in silicon wafers. However, plasmonic substrates have also been demonstrated using metallic-coated resist patterns [22,23,24], which can further reduce the time and cost of fabricating SERS substrates by avoiding the need for silicon etching. During the NIL process, a prefabricated NIL stamp with a designed pattern is pressed into a polymer resist coated on a substrate and then illuminated with UV light (UV-NIL) or heated (thermal NIL) to cure the resist. The NIL stamp for submicron patterns requires an EBL fabricated master template (DUV can also be used [25]); however, these stamps can be used multiple times. By using NIL fabrication, the time-consuming EBL process only needs to be performed once, after which replications can be made at high throughput and low cost. Furthermore, the size and shape of nanostructures produced by NIL are homogenous and reproducible on large pattern areas [10]. An additional advantage of the NIL technique is that it can handle certain topographies on the substrate through a soft stamp. In this work, we utilized this UV-NIL technique to fabricate our novel C-shaped design.

In this paper, we present the design, fabrication, and characterization of SERS substrates using UV-NIL resist patterns covered by gold evaporation. A C-shape pattern was simulated and optimized for a 785 nm excitation wavelength using the finite difference time domain (FDTD) method, which is discussed in detail in Section 2. Section 3 covers the fabrication details of the SERS platforms. The functionality of the SERS substrates was experimentally verified by measuring SERS spectra of Rhodamine 6G, which is commonly used to characterize SERS platforms in many reported studies. The results are presented in Section 4.

## 2. Design

The shape and size of the nanostructure pattern were designed to obtain plasmon resonance in the 700–900 nm wavelength range, with the intention to create multiple hotspots in the electrical field and take fabrication limitations into account. Thus, shapes that require less sharp or pointed edges, such as C-shaped nanostructures, are preferred. The simulations were carried out using the FDTD method implementation in the Ansys Lumerical version R1.4 commercial software. The nanostructured pattern investigated in this paper is an array of C-shape in nanostructures arranged in a lattice, as illustrated in Figure 1a. Full 3D substrates were modelled: an array of C-shaped nanostructures (height 200 nm, width 40 nm and 50 nm, and diameter 300 nm) made of resist, positioned on a silicon (Si) substrate, coated with a gold layer (30–50 nm) (Figure 1b). The refractive index of the gold and silicon substrate used in the simulations was from Palik et al. [26], which was included in the FDTD materials database, while the refractive index of the NIL resist was measured using the M-2000 Ellipsometer (J.A. Woollam, Lincoln, NE, USA) and it was 1.54 at 1000 nm. Three C-shaped structures were included in a rectangular simulation volume, with the perfectly matched layer (PML) boundary conditions in the X direction (the propagation direction) and periodic boundary conditions in the Y and Z directions. The substrate was illuminated from the top by a broadband plane wave, with a 500–1600 nm wavelength range. The reflected intensity was measured above the substrate, while the electric field distribution was recorded at the gold–air interface.

Firstly, the reflection was simulated for two widths (W) of the resist C-shape elements: 40 and 50 nm, while the length (L) was kept constant at 300 nm (Figure 1a). The height of the resist nanopattern was 200 nm, the height of the resist below the nanopattern was 70 nm, and the thickness of the gold layer was 30 nm. Reflection showed multiple resonances in the 500–1600 nm wavelength range (Figure 2c). To ensure maximum SERS enhancement, the plasmon resonance should be located slightly above the excitation laser wavelength, enabling the enhancement of both the laser and the scattered Raman-shifted wavelengths [27,28]. Since our selected Raman excitation wavelength was 785 nm, and the sample selected for testing was Rhodamine 6G (R6G), the relevant wavelengths for the resonance were in the 785–900 nm range. However, it is important to note that several samples have Raman peaks in a similar wavelength range, such as glucose and pesticides [28,29], when excited with 785 nm. The reflection spectra are presented in Figure 2c, showing multiple resonances in the visible and infrared wavelengths. The electric field concentrated at the outer edges and tips of the C-shape elements, showing two main distinct resonances that appeared for different wavelengths (Figure 2a,b), which may be due to the distance between the C-shaped particles being different within and between rows. For the 50 nm wide C-shape, the resonance shifted to shorter wavelengths in the visible range. Moreover, the plasmon resonance could be shifted by changing the width of the gold layer film. For the 40 nm C-shape width, the reflection was simulated for four different Au widths: 70 nm, 80 nm, 90 nm, and 100 nm, giving a shifted plasmon resonances at 850 nm, 790 nm, 760 nm, and 735 nm, respectively (Figure 2e,f). The origin of these resonance shifts can intuitively be understood given the changes in the geometry: For instance, a thicker gold layer means that the relative penetration depth of the electric field into the polymer layer beneath was affected (the skin thickness of Au was of the same order of magnitude as the coating widths used here in the wavelength region). In addition, scaling dimensions of resonators are known to lead to resonance shifts. Therefore, while the lateral dimensions of the patterns were fixed according to the EBL patterned master template, the plasmon resonances could be modified by varying the deposited Au thickness.

It should be noted that the asymmetry of C-shaped structures can result in a polarization-dependent response, potentially offering higher enhancement for a specific polarization. However, for this study, we did not account for polarization and opted to use a nonpolarized source instead.

## 3. Fabrication

The UV-NIL process used in this work uses a soft stamp and consists of three steps (Figure 3). Firstly, the master template was fabricated, which had the design pattern etched into the Si wafer. Here, the master was purchased from NIL Technology (NILT, Kongens Lyngby, Denmark) according to our specified designs. It was fabricated using EBL and a lift-off process. Secondly, the soft stamp was fabricated by spin-coating the NIL master with the silicone-based stamp material GMN-PS90 (OpTool, Veberöd, Sweden). The stamp material was then transferred onto a plastic carrier foil by roll-on, and curing was carried out via UV-exposure conducted on an EVG620 Smart NIL system (EVG, St. Florian am Inn, Austria). Second, the fabricated stamp, which was an inverted copy of the master, was used to imprint the nanopattern into a photoresist on a silicon wafer (wafer thickness 575 µm). The wafer was primed with adhesion promoter mr-APS1 (Micro Resist Technology, Berlin, Germany), and spin-coated with resist mr-NIL210–200 nm at 4000 RPM for 60 s, giving the photoresist a 170 nm thickness. Finally, roll-on imprinting using a EVG620 Smart NIL was performed, and the imprint was exposed with UV using 31 mW/cm^2^ for 130 s. Finally, the nanostructures imprinted in the resist were coated with gold by evaporation using a sputter tool Q150R ES (Quorum Technologies, UK), yielding a final width W of the C-shape of around 70 nm.

The fabricated substrates were characterized using a scanning electron microscope (SEM) (Figure 4), showing an excellent homogeneity over the whole chip area (1000 × 1000 µm). The shape of the C-shaped nanopattern was very well reproduced when compared to the Si master (Figure 4a,b). Both in the master and the UV-NIL imprints, the ends of the C-shape elements were more rounded when compared with the original design (Figure 1a). The rounding of the structures was taken into account in the simulations by adding rounded edges to the patterns. Figure 4c,d show the cross-section image of the UV-NIL imprinted structures. The height of the structures was 220 nm and the Au coverage/distribution is visible. The C-shape elements were observed to be well covered with Au; however, the gold layer was thicker on top of the structures. 

## 4. Experimental Setup

To conduct SERS measurements, we used an i-Raman Plus spectrometer (BWTEK) with a fiber optic probe and a 785 nm excitation wavelength. The fiber probe was mounted on an xyz translation stage to optimize its position above the substrate (Figure 5). All measurements were conducted using a laser power of 68 mW and a measurement time of 10 s. For the sample, we used Rhodamine 6G (R6G, Merck KGaA, Darmstadt, Germany) which was diluted in water.

## 5. Results and Discussion

For the measurement leading to Figure 6a, we first pipetted a 3 µL droplet of the R6G sample with a concentration of 10 µM onto the fabricated SERS substrate and measured the Raman signal (blue curve). The sample was then left to dry for 20 min, and we measured the Raman signal again (red curve). The dried sample exhibited a much higher intensity of R6G peaks, as the R6G molecules were presumably closer to the SERS substrate, leading to a stronger Raman signal enhancement. However, both dried and liquid samples were measured efficiently. To estimate the enhancement factor (EF) of the fabricated substrates, we measured a sample of dried R6G with a 1 mM concentration on a gold-coated Si substrate without a nanopattern and the R6G peaks had an extremely low intensity in the measured Raman spectra (green curve, in Figure 6a). *EF* was calculated for the 1361 cm^−1^ peak using the following equation [30]:$$EF=\frac{I_{SERS}\cdot C_{ref}}{I_{ref}\cdot C_{SERS}}$$

The terms *I_SERS_* and *I_ref_* in the equation represent the measured Raman intensities of the analyte (R6G) on the fabricated SERS substrate and on the gold-coated Si substrate, respectively, which serve as a reference after background subtraction. The terms *C_SERS_* and *C_ref_* represent the corresponding concentrations of the analyte (R6G) used in the measurement.

For Figure 6a, the measured Raman intensity on the fabricated SERS platform (*I_SERS_*) for the 1361 cm^−1^ Raman peak was 1.66 × 10^4^ and 1.95 × 10^3^ for the dried and liquid analyte (R6G), respectively. The measured Raman intensity from the reference substrate, i.e., the gold-coated Si substrate (*I_ref_*), was 68. The concentrations of the analyte (R6G) used for the fabricated SERS platform and the reference platform, i.e., the gold-coated Si substrate, were 10 µM and 1 mM, respectively. The enhancement factor (*EF*) for the dried R6G was 2.4 × 10^4^ and 2.8 × 10^3^ for the liquid R6G sample.

To demonstrate the reusability and stability of our platform, the SERS substrate was immersed in IPA for 10 min, dried, and then the SERS signal was measured to make sure the R6G was washed away. Afterward, the R6G sample was reapplied and it was measured again after drying. The cycle of washing and reapplying R6G was repeated three times over the course of a week. Each time, the SERS spectra were measured in five different locations on the chip, in the corners, and the center of the chip. A baseline correction was applied to the collected SERS spectra using a second derivative technique and range independent baseline correction algorithm discussed in Reference [31]. The SERS signal measured in five different locations of the substrate shows very good uniformity over the whole chip area (Figure 6b). The relative standard deviation (RSD) values of the Raman intensity at the Raman peaks in the 1100–1600 cm^−1^ range were less than 4.3%. The RSD for the 1361 cm^−1^ peak for all three series of measurements was 8.4%. The higher RSD values for the measurement’s series were thought to be related to sample drying rather than the SERS substrate.

## 6. Conclusions

We present the design and fabrication of nanopatterned SERS substrates fabricated using the UV-NIL technique. The optical properties of the C-shape design can be easily tuned by changing the thickness of the deposited gold layer and, therefore, different wavelength ranges can be enhanced. The fabricated substrates show good uniformity over the whole chip area (1000 × 1000 µm) and the process of using a UV-NIL resist coated with gold as SERS substrates significantly reduces the time and cost of fabrication. The SERS measurement show a good and uniform enhancement factor for both the liquid and dried samples and great stability over multiple measurements and cleaning cycles. Going forward, our platform could be integrated with microfluidics to enable continuous detection, which is important for practical applications in medical and environmental sensing.

## Figures and Tables

**Figure 1 nanomaterials-13-01598-f001:**
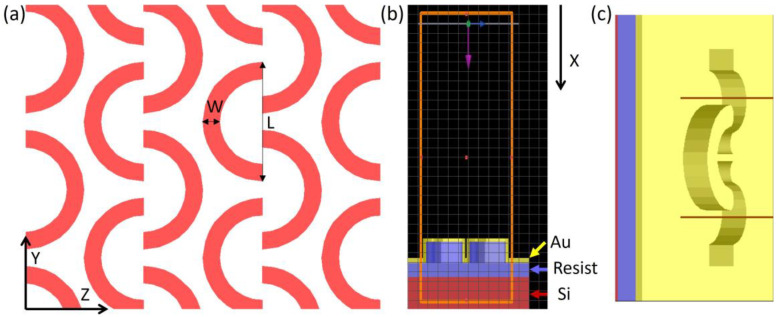
(**a**) Design of the C-shape nanopattern; width W = 40 nm and 50 nm, length L = 300 nm. (**b**) Cross-section of the simulation model: red part represents the Si substrate, blue represents the resist pattern, and yellow represents the gold layer. The pink arrow indicates the direction of the light propagation (X). (**c**) Top view of the simulation window.

**Figure 2 nanomaterials-13-01598-f002:**
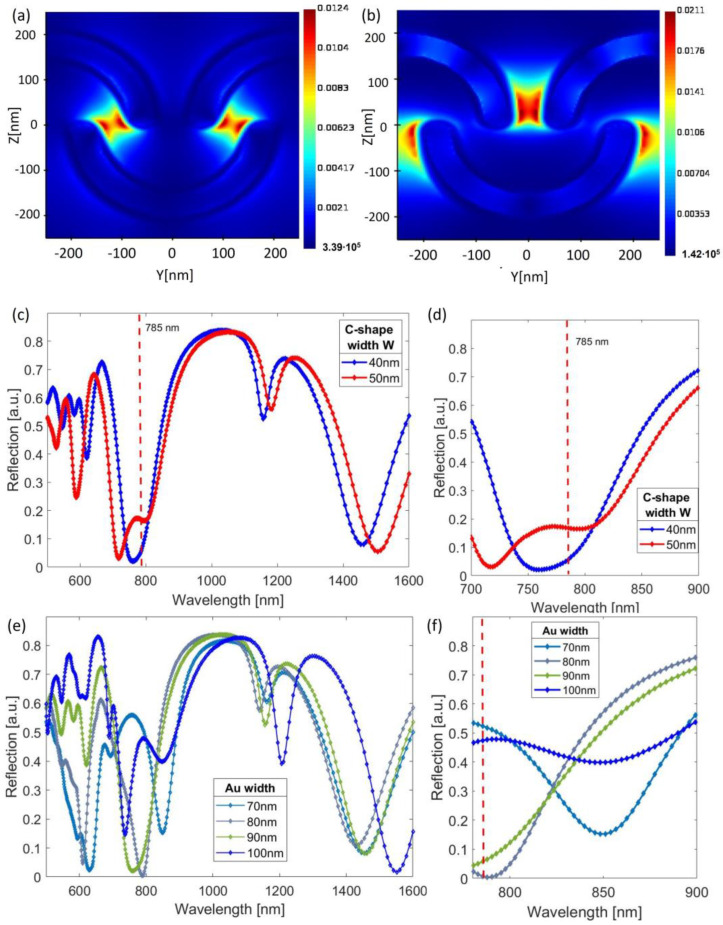
The simulated electric field distribution was analyzed for two different wavelengths: (**a**) 800 nm and (**b**) 1100 nm. The reflection spectra were simulated for C-shaped resist particles with widths of 40 nm and 50 nm, as shown in panel (**c**) and in the zoomed-in section in (**d**). Panel (**e**) shows reflection spectra for the C-shaped resist particle with 40 nm width and different Au thickness (30–50 nm), while (**f**) shows the zoomed-in area of panel (**e**).

**Figure 3 nanomaterials-13-01598-f003:**
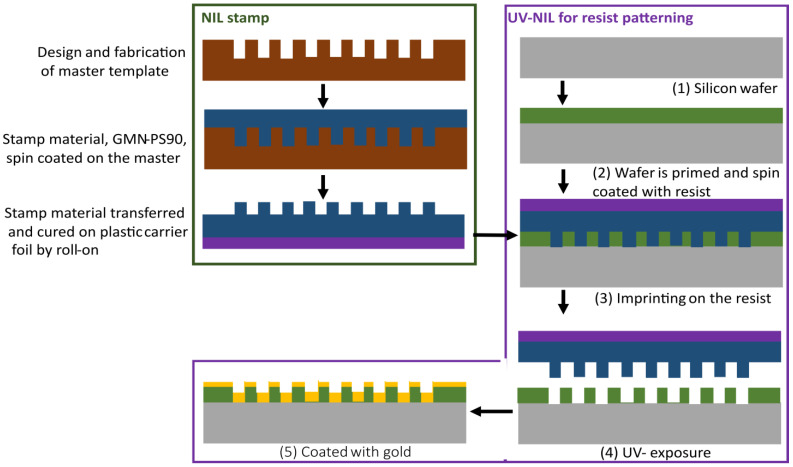
UV-NIL process flow used for the fabrication of SERS substrates.

**Figure 4 nanomaterials-13-01598-f004:**
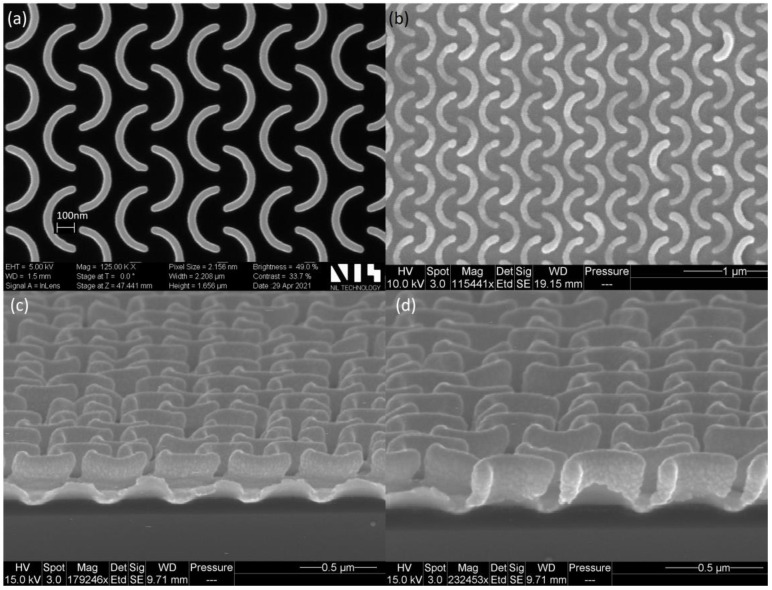
SEM image of the C-shape nanopattern: (**a**) Top view of the Si master (NIL Technology ApS, Denmark); (**b**) top view of the UV-NIL fabricated SERS pattern coated with Au; (**c**,**d**) cross-section of the UV-NIL SERS substrate.

**Figure 5 nanomaterials-13-01598-f005:**
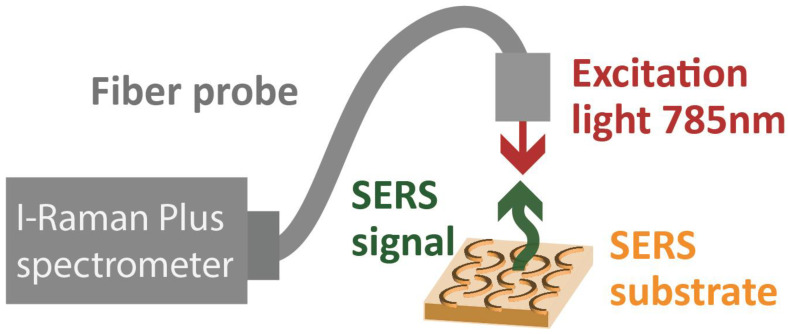
Experimental setup: i-Raman Plus spectrometer (BWTEK) with a fiber optic probe fixed on top of the SERS substrate.

**Figure 6 nanomaterials-13-01598-f006:**
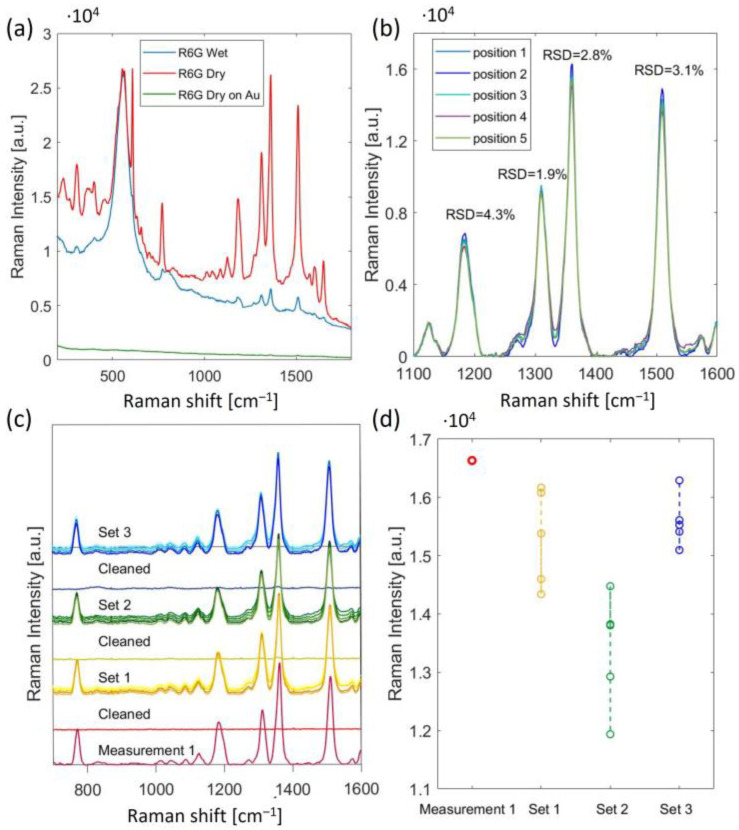
(**a**) SERS spectra of 10µM of R6G measured on the fabricated SERS: wet sample (blue); dried sample (red); 1 mM of R6G dried on Au-coated Si (green). The broad peak observed around 500 cm^−1^ is likely attributed to the resist polymer; (**b**) SERS spectra of dried 10 µM of R6G measured in 5 different positions of the chip; (**c**) reusability and stability test for four detection and cleaning cycles; (**d**) Raman intensity for the 1361 cm^−1^ peak for each measurement series.

## Data Availability

The data supporting the findings of this study can be made available by the corresponding authors upon a reasonable request.

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
