# Peer review of "UV-Nanoimprint Lithography for Predefined SERS Nanopatterns Which Are Reproducible at Low Cost and High Throughput"

_nanomaterials, 2023, doi:10.3390/nano13101598_

Round 1
Reviewer 1 Report
In this article, the authors present exciting results on the SERS measurements using substrates fabricated using inexpensive nano imprint lithography technique. The article will be a good reference for researchers working on the topic. Reviewer recommends the publication of the article with the following suggestions.
1. If there are any formulation which define the correlation between the dimensions/shape of the and patterns and the ERS single please include it
2. Why the Rhodamine 6G was selected for the SERS studies? Please discuss the general applicability of your substrate (nanopatterns) and specify if there are any surface specific interactions which can limit the detection of other molecules on your substrates.
3. Fig. 2 caption is confusing please rewrite it clearly. Fig. 2 q0, 2b) recommended to add the color bar to the figures, axis is not readable.
4. Fig. 5 check the double captions.
5. Line 174, please refer to the correct figure, line 185 please define all the terms of the equation.
6. Please define all abbreviations eg. PML line 85 in the article and check spelling eg. line 89
Please checkk the spellings
Reviewer 2 Report
The authors present the design, fabrication, and characterization of SERS substrates using UV-NIL resist patterns covered by gold evaporation. Technically, the reported results are solid and comprehensive, which is suitable for the publication in Nanomaterials.
To improve the quality of the manuscript, the authors should also show sufficient supporting for clarifying the theme of the manuscript. In order to further support its publication, the authors should also address the following points, as listed.
1. In this work, the authors used C-shape nanopattern to realize SERS. The asymmetric C shapes are typically polarization sensitive. To realize a full range of SERS, the authors should discuss the polarization-dependent SERS effects.
2. The author should also clarify the selection of the nanopattern for SERS applications, such as the advantages of using the C-shape pattern to demonstrate the topic of this research work.
3. The authors should also clarify more about the stability of their system using the resist as the supporting layer. Polymer is easy to show Raman signals. It means that the SERS signal will always contain a background of the materials using in the fabricated SERS sample. The authors should support that claim with experimental data, if possible.
4. There are more works using the same technical process to fabricated the nanoimprinted substrates using in the other field, which should be cited properly.
a. Master origination by 248 nm DUV lithography for plasmonic color generation. Applied Physics Letters 118(14):141103, 2021. DOI: 10.1063/5.0046163
b. Nanoimprinted SERS-Active Substrates with Tunable Surface Plasmon Resonances. J. Phys. Chem. C 2007, 111, 18, 6720–6723.
Minor editing of English language required.
Reviewer 3 Report
The manuscript presents the design, fabrication, and characterization of nanopatterned Surface Enhanced Raman Spectroscopy (SERS) substrates using the UV-Nanoimprint Lithography (UV-NIL) technique. The authors employ a C-shaped nanopattern design, which allows for tunable optical properties by changing the thickness of the deposited gold layer. The research content of this paper is interesting, and the results have a certain reporting value. The text has a fair amount of length and an acceptable structure. Nonetheless, there are still some issues in this manuscript that need to be seriously addressed. For these reasons, I can only agree to publish this manuscript on Nanomaterials after major revisions.
The following issues should be appropriately improved and further elaboration:
1. In Section 3, Fabrication, the authors describe the fabrication process of the SERS substrates using UV-NIL technique. However, the discussion on the advantages and disadvantages of this technique compared to other fabrication methods (such as Electron Beam Lithography) is not provided. It would be helpful to have a comprehensive comparison that highlights the benefits of using UV-NIL in terms of fabrication time, cost, and scalability.
2. In Section 5, Results and Discussion, the authors present the Enhancement Factor (EF) calculation for the fabricated substrates. However, it is unclear how the values for ISERS and Iref were obtained. Please provide a more detailed explanation of the measurement process and the calculations involved in determining these values.
3. The manuscript lacks a thorough statistical analysis of the results. In Figure 6b, the authors present the uniformity of the SERS signal measured in five different locations on the substrate. It would be valuable to include a more in-depth statistical analysis, such as the calculation of the coefficient of variation or other relevant parameters, to better quantify the uniformity and reproducibility of the measurements.
4. The manuscript mentions the potential integration of the SERS substrates with microfluidics for continuous detection in medical and environmental sensing applications. However, this point is not explored in depth. Please provide a more detailed discussion on how the substrates could be incorporated into microfluidic devices and the possible challenges and advantages of such integration.
5. The authors mention the tunability of the optical properties by changing the thickness of the deposited gold layer. Please provide a more detailed discussion on the impact of gold layer thickness on the plasmonic properties of the C-shaped structures, supported by numerical simulations or experimental data.
Minor editing of English language required.
Reviewer 4 Report
1. Suggest doing a characterization on the dimensional accuracy analysis on the fabricated nanopattern.
2. Suggest explaining each term in equation found in section 5. ?? = ????? ∙ ???? /???? ∙ ????
3. It was mentioned that Isers was 1.66·104 . There's no unit shown. Should there be a unit for this value?
4. In the conclusion section, can the authors also suggest potential future research and application of this work?
5. In the introduction, the authors should also discuss about other fabrication techniques such as 3D electronic printing using aerosol jet and inkjet printing and discuss how the proposed technique is more suitable compared to these techniques. Suggest discussing and citing the following papers:
1. G. L. Goh, S. Agarwala, and W. Y. Yeong: 'High Resolution Aerosol Jet Printing of Conductive Ink for Stretchable Electronics', Proceedings of the 3rd International Conference on Progress in Additive Manufacturing (PRO-AM), Nanyang Technological University, Singapore, 2018, 109-114.
2. (2016). Inkjet‐Assisted Nanotransfer Printing for Large‐Scale Integrated Nanopatterns of Various Single‐Crystal Organic Materials. Advanced Materials, 28(15), 2874-2880.
3. (2021). Three-dimensional nanoprinting via charged aerosol jets. Nature, 592(7852), 54-59.
6. Can the proposed technique be used to fabricate pattern on curved surfaces? Suggest discussing the limitation and potential of the proposed technique for conformal fabrication of nanopatterns on intricate surfaces. Suggest comparing it with other fabrication techniques like 3D electronic printing techniques. Suggest citing and discussing the relevant work below.
a. (2019). Assembly and applications of 3D conformal electronics on curvilinear surfaces. Materials Horizons, 6(4), 642-683.
b. Fabrication of design-optimized multifunctional safety cage with conformal circuits for drone using hybrid 3D printing technology. Int J Adv Manuf Technol 120, 2573–2586 (2022). https://doi.org/10.1007/s00170-022-08831-y
c. . (2022). Review on 3D fabrication at nanoscale. Autex Research Journal.
Nil
